# Peroxiporins and Oxidative Stress: Promising Targets to Tackle Inflammation and Cancer

**DOI:** 10.3390/ijms25158381

**Published:** 2024-08-01

**Authors:** Inês V. da Silva, Monika Mlinarić, Ana Rita Lourenço, Olivia Pérez-Garcia, Ana Čipak Gašparović, Graça Soveral

**Affiliations:** 1Research Institute for Medicines (iMed.ULisboa), Faculty of Pharmacy, Universidade de Lisboa, 1649-003 Lisboa, Portugal; 2Department of Pharmaceutical Sciences and Medicines, Faculty of Pharmacy, Universidade de Lisboa, 1649-003 Lisboa, Portugal; 3Division of Molecular Medicine, Ruđer Bošković Institute, 10000 Zagreb, Croatia

**Keywords:** aquaporin, hydrogen peroxide, membrane transport, redox signaling

## Abstract

Peroxiporins are a specialized subset of aquaporins, which are integral membrane proteins primarily known for facilitating water transport across cell membranes. In addition to the classical water transport function, peroxiporins have the unique capability to transport hydrogen peroxide (H_2_O_2_), a reactive oxygen species involved in various cellular signaling pathways and regulation of oxidative stress responses. The regulation of H_2_O_2_ levels is crucial for maintaining cellular homeostasis, and peroxiporins play a significant role in this process by modulating its intracellular and extracellular concentrations. This ability to facilitate the passage of H_2_O_2_ positions peroxiporins as key players in redox biology and cellular signaling, with implications for understanding and treating various diseases linked to oxidative stress and inflammation. This review provides updated information on the physiological roles of peroxiporins and their implications in disease, emphasizing their potential as novel biomarkers and drug targets in conditions where they are dysregulated, such as inflammation and cancer.

## 1. Introduction

Aquaporins are a family of integral membrane proteins facilitating water and small polar molecule permeation across membranes, driven by osmotic or solute gradients [1,2]. These highly selective channels are vital in maintaining water balance in tissues and organs and are found in various organisms, including animals, plants, and microorganisms.

AQPs are structured in membranes as homotetramers, consisting of four identical monomers with a molecular weight of around 28 kDa, each monomer acting as an independent pore. The monomer is formed by six highly hydrophobic transmembrane domains and two half-helices containing asparagine–proline–alanine (NPA) motifs that are the signature of the AQP family, and N-terminal and C-terminal sequences facing the cytosol. The aquaporin channel achieves selectivity through two filters within the pore: (i) a size-selective filter made of aromatic/arginine (ar/R) residues near the extracellular entrance, which dictates the size of molecules that can pass (2.8 Å for classical aquaporins and 3.4 Å for aquaglyceroporins), and (ii) a charge-selective filter composed of two highly conserved NPA motifs that act as dipoles, preventing ion passage through the channel (Figure 1A). Additionally, a central pore located in the middle of the tetramer has been proposed in some classes of AQPs to serve as a pathway for the flux of gases or charged particles, such as ions [3].

The AQP homologues identified in humans are grouped according to their primary structure and selectivity into three main groups: (i) classical or orthodox AQPs (AQP0, 1, 2, 4, 5, 6, 8) are considered primary water channels, although AQP6 and AQP8 also transport anions and ammonia; (ii) aquaglyceroporins (AQP3, 7, 9, 10), which also permeate small non-charged molecules such as urea and glycerol; and (iii) unorthodox or S-aquaporins (AQP11 and 12), named superaquaporins due to the very low homology with other AQP subfamilies, which can transport water and glycerol across intracellular membranes [1,4,5,6]. However, the molecular similarity of hydrogen peroxide (H_2_O_2_) with water suggested that at least some AQPs could simultaneously facilitate H_2_O_2_ passage. The permeability to H_2_O_2_ emerged as a feature of a new sub-group named peroxiporins (AQP0, 1, 3, 5, 6, 7, 8, 9, 11) [7,8,9,10,11,12,13,14,15,16,17] which is being progressively expanded as new peroxiporins are being identified (Figure 1).

Experimental and computational studies have made efforts to unveil the molecular basis of H_2_O_2_ conductance and regulation through AQPs. The pore width of each AQP homologue at the ar/R region may explain, at least in part, their ability to accommodate the H_2_O_2_ molecule, facilitating permeation [11]. Moreover, AQPs with higher water permeability are expected to display higher H_2_O_2_ conductance. The combination of the widened selectivity filter with a larger rate of channel permeability may dictate the ability to permeate H_2_O_2_ in addition to water. Nonetheless, the residue arrangement at the vicinity of the selectivity filter and channel conformation may favor H_2_O_2_ interactions to form hydrogen bonds and facilitate diffusion [11,18]. However, the transport specificities of two plant aquaporins (AtPIP2;4, SoPIP2;1) compared with human AQP1 show that despite equal efficiency for water transport of the three homologues, the plant AQPs show higher efficiency for H_2_O_2_, which could not be explained by the monomers’ structural differences [19].

Importantly, a reversible gating mechanism that involves the persulfidation of a critical cysteine residue in the channel (C53) was identified for AQP8, which may explain the possible control of H_2_O_2_ gradients caused by extracellular NADPH oxidases (NOX) that release H_2_O_2_ into the extracellular space upon activation of tyrosine kinase receptors [20]. Treatment with H_2_S (primarily produced by cystathionine β-synthase) induced the blockage of H_2_O_2_ influx in unstressed cells. Molecular modeling indicates that the persulfidation of C53 may influence the peroxiporin conductance in several ways. It could cause steric hindrance with the aromatic ring of the nearby histidine located at the narrowest part of the channel (H72) (Figure 1C,D) or disrupt the hydrogen bonds formed by this histidine with H_2_O_2_, which are essential for its transport through the pore. It is reasonable to assume that H_2_O_2_ molecules, when transported through AQP8, promote C53 sulfenylation, making it susceptible to the H_2_S primarily produced by cystathionine β-synthase (CBS). This mechanism can control H_2_O_2_ permeation across membranes, reducing cellular oxidative stress and potentially regulating downstream signaling pathways. A predictive structure of AQP8 and the proposed mechanism of regulation by H_2_O_2_ are represented in Figure 1.

Oxidative stress is both a cause and consequence of cancer development. DNA damage resulting from reactions with oxidizing agents can lead to mutations and the dysregulation of key molecules that control cell proliferation [21]. During cancer progression, cells undergo metabolic changes due to mutations, rapid growth, and alterations in the microenvironment, such as hypoxia and nutrient deprivation. These changes influence the redox balance within the cell, creating a vicious cycle where the overexpression of H_2_O_2_-producing enzymes increases intracellular H_2_O_2_ levels, which in turn elevates the antioxidative defense mechanisms. H_2_O_2_ stimulates signaling pathways involved in cell proliferation and pathways promoting its detoxification, allowing cancer cells to regulate H_2_O_2_ levels effectively. Enhanced antioxidative protection also provides a defense against cancer therapies, adding another layer of complexity to treatment.

In this overview, we examine the role of peroxiporins—membrane channels that regulate H_2_O_2_ flux—and their impact on oxidative stress as a promising target for controlling cancer growth. By modulating peroxiporin activity, it may be possible to disrupt the oxidative balance in cancer cells, inhibit tumor growth, and enhance the efficacy of existing treatments. This review underscores the potential of targeting peroxiporins in the development of innovative anti-cancer therapies.

## 2. Peroxiporins in Redox Homeostasis

Oxygen and oxidative metabolism have been shown to be strategic advantages for the organisms that adopted this energy-generating pathway. On the other hand, oxygen is also toxic, and its toxicity is achieved through the formation of reactive oxygen species (ROS). ROS are highly reactive molecules and free radicals, such as superoxide anion (•O_2_^−^), hydroxyl radical (OH), and H_2_O_2_, which easily oxidize biological molecules, thereby changing their activity and function [22]. ROS levels are augmented in stress conditions by numerous exogenous factors. Interestingly, ROS are also produced endogenously, either by imperfections in cellular processes or to support certain processes. Yet, studies imply that endogenous ROS production is not due to imperfections; rather, these ROS actually play a role in the processes that stimulate their production. Electron leakage from mitochondrial complex I is such a process. ROS, mainly •O_2_^−^ and H_2_O_2_, formed by complex I are quenched by the redox-sensitive cysteines of signaling molecules, thereby modifying the cellular response to O_2_ levels [23]. Moreover, since H_2_O_2_ prompts reversible modifications of thiol groups, whereas •O_2_^−^ prompts irreversible modifications [24], the dismutation of •O_2_^−^ by superoxide dismutase (SOD) also controls the signaling intensity and pathways. By regulating the level of these ROS, the antioxidative defense system provides fine-tuning of signaling [23]. Another example of controlled ROS production is the activity of NOX, in which NOX1 to NOX5, DUOX1, and DUOX2 produce ROS as their primary function to support cellular processes and reactions [25]. Interestingly, although the NOX protein family produces H_2_O_2_, each member has specific and distinctive roles in cellular signaling due to their subcellular and organ-specific localization [26]. H_2_O_2_ produced by NOX4 is transported into the cells by peroxiporins (AQP3 and AQP8), crucial players in redox signaling [12]. Additionally, NOX2 is coupled with AQP3 in the plasma membrane, creating a local increase in H_2_O_2_ that supports epidermal growth factor/receptor (EGF/EGFR) signaling [27]. Further, co-immunoprecipitation experiments showed that in addition to AQP3, NOX2 interacts with AQP5 and AQP8 [28]. H_2_O_2_ produced by NOX2 enters the cell via peroxiporins and supports Fenton’s reaction, lipid peroxidation, and results in ferroptosis [28]. Another source of ROS is the endoplasmic reticulum (ER), where H_2_O_2_ is used for protein folding. The above-mentioned NOX4 is localized in the ER, although it can be found in mitochondria and the nucleus [29]. Here, as a source of H_2_O_2_, NOX4 contributes to the regulation/oxidation of PTP1B and thus the regulation of EGFR signaling [30] and, to a much lesser extent, to oxidative protein folding [31]. Oxidative protein folding is rather dependent on endoplasmic reticulum oxidoreductin 1α (ERO1α). When ERO1α is downregulated, the resulting need for H_2_O_2_ is compensated by its influx from the mitochondria through AQP11 [31].

As ROS can be detrimental to the surrounding molecules and the cell, the intracellular levels of ROS are controlled to maintain their positive effects and prevent side effects. Therefore, cells developed complex defense mechanisms, either nonspecific or specific, but also intertwining and able to compensate for the others’ deficiencies [32]. The first line of defense is composed of small, non-specific scavengers such as vitamin C and vitamin E, glutathione (GSH), thioredoxin (Trx), ubiquitin, transferrin, ferritin, bilirubin, α-lipoic acid, melatonin, melanin, uric acid, and β-carotene [22]. The next line of defense, which also uses small molecules (e.g., GSH and Trx), are enzymatic cascades that lead to the complete detoxification of ROS. The GSH and Trx systems are thiol-based systems that react to the wide selection of electrophiles, neutralizing them completely. These systems are recycled through the activity of their specific peroxidases and reductases to restore the initial state of the thiols and use the NADPH generated by the pentose-phosphate pathway [33]. The first enzymatic cascade leading to the complete detoxification of specific ROS, namely •O_2_^−^, includes SOD1, SOD2, and SOD3 localized in the cytoplasm, mitochondria, and extracellular matrix, respectively, generating H_2_O_2_ which is further neutralized by catalase (CAT) to water and oxygen [34]. In addition to CAT, other enzymes can neutralize H_2_O_2_, such as glutathione peroxidases (GPX) and peroxiredoxins (Prx). The synthesis of these small peptides and enzymes is regulated by redox-sensitive antioxidant transcription factors (Figure 2).

To maintain redox homeostasis, cells have a network of signaling pathways and transcription factors that coordinate the cellular response to oxidative stress. Redox-responsive signaling pathways are usually localized near sources of ROS generation where they are activated and promote cell survival and metabolic adaptation [35]. The overexpression of ROS-producing proteins such as NOX [36] is found in several types of cancer [37]. This overexpression leads to increased ROS levels, which activate signaling pathways that contribute to cancer proliferation, thereby supporting various hallmarks of cancer (reviewed in [37,38]). Simultaneously, elevated ROS levels in cancer cells trigger antioxidative responses, including cell cycle arrest, DNA repair, apoptosis, and the activation of redox-responsive transcription factors, thereby enhancing the cancer cells’ defense mechanisms against ROS.

The activation of a transcription factor must be strictly modulated even in the state of oxidative stress. It is proposed that different transcription factors respond to ROS levels in a concentration- and/or time-dependent manner. NRF2 (Nuclear Factor Erythroid 2-Related Factor 2) is the first-line defense transcription factor, responding to low and moderate ROS levels; higher levels are needed to activate AP-1 (Activator Protein 1) and NF-κB (Nuclear factor kappa B), along with FOXO (Forkhead box O) and HIF-1 (Hypoxia-Inducible Factor-1); finally, as the last response, there is the activation of TP53 (Tumor Protein P53) [39]. This hierarchical defense strategy ensures a controlled response to variable levels of ROS within the cell, preventing the simultaneous activation of all transcription factors. Additionally, different transcription factors are activated in different ways, and the activation of a transcription factor can occur by controlling its synthesis and degradation or by controlling the activity of a preexisting transcription factor [40]. Surely, other transcription factors are involved, directly or indirectly, in the cellular response to oxidative stress that are not mentioned here, but their detailed discussion falls out of the scope of this paper. It is worth mentioning that the role of a major antioxidant transcription factor, NRF2, as a first responder to cellular ROS is achieved through KEAP1 (Kelch-like ECH-associated protein 1). The KEAP1-dependent degradation via ubiquitination and proteasomal degradation regulates NRF2 activity in unstressed conditions. In the presence of ROS or electrophiles, redox-sensitive cysteine residues on KEAP1 are modified, restricting its activity. Subsequently, newly synthesized NRF2 directly translocates to the nucleus where it heterodimerizes with small MAF proteins and binds to antioxidant-responsive elements (AREs) within the regulatory region of multiple antioxidant genes [41], such as NAD(P)H quinone oxidoreductase (NQO1), heme oxygenase (HO), Trx, Prx, GPX, and many others [42]. The constitutive transcription and fast degradation of NRF2, in combination with the highly sensitive ROS sensing by KEAP1, guarantees a rapid response, even in the presence of low levels of ROS, making NRF2 in charge of the initial cellular response to ROS.

Another important redox-sensitive transcription factor, responding to higher levels of ROS, is a group of FOXO transcription factors that act as homeostasis regulators in metabolic and oxidative stress [43]. They regulate cell cycle arrest, DNA repair, apoptosis, and importantly, antioxidant protection by upregulating CAT and SOD2 [44]. Depending on the ROS source and type of modification, FOXO transcription factors can be activated or inactivated. They can be modulated through the regulation of cytoplasmic/nuclear shuttling, gene transcription, mRNA stability, and different posttranslational modifications (phosphorylation, methylation, acetylation, O-glycosylation, and ubiquitination) [44]. This complex regulation of FOXO transcription factors, often called the FOXO code [45], enables additional fine-tuning of antioxidant reactions in the cell.

## 3. Peroxiporins in Oxidative Stress and Signaling

Redox signaling represents one of the fundamental regulatory pathways in the physiology of cells and organisms. H_2_O_2_, the most common ROS in cells, acts as a secondary messenger. The perturbation of physiologic H_2_O_2_ cellular content increases cellular oxidative stress, a feature implicated in aging and pathological conditions such as inflammation and tumorigenesis [46,47]. Due to their function as peroxiporins and their role in oxidative stress, certain aquaporin homologs have been implicated in the onset of several disease conditions. Examples of such implications are detailed below.

AQP3 peroxiporin activity is crucial for the stimulation of NF-κB signaling in keratinocytes and in the development of psoriasis [48]. In keratinocytes, NOX2 generates H_2_O_2_ in response to TNFα, which in turn is transported into the cytosol via AQP3. Increased intracellular H_2_O_2_ regulates protein phosphatase 2A activity, activating the NF-κB signaling pathway and influencing the etiology of the disease [48]. Moreover, IL-23-induced psoriasis in the AQP3 knockout (KO) murine model is less aggressive than in WT mice and is accompanied by impaired NF-κB activation and low intracellular H_2_O_2_ accumulation. AQP3 was also reported to play an important role in the etiology of Vitiligo [49,50]. Using primary cells derived from lesional and non-lesional skin biopsies of Vitiligo patients, AQP3 expression was shown to negatively correlate with the progression of the disease, along with NRF2 and NQO1. AQP3-silenced human primary keratinocytes showed similar results to lesional biopsies, and NRF2 was further reduced upon H_2_O_2_ treatment. Moreover, ROS-rich conditioned media from AQP3-silenced keratinocytes treated with H_2_O_2_ induced human primary melanocyte apoptosis, suggesting that in Vitiligo, apoptotic melanocytes are a result of AQP3 deficiency in keratinocytes [49]. These reports highlight the key role of AQP3 peroxiporin in skin pathophysiology.

AQP11 is an important protein in maintaining endoplasmic reticulum physiological function and has been detected in high levels in kidney proximal tubule epithelial cells. The first segment of the proximal tubule stains strongly positive for AQP11 and is characterized by an intense glucose flux. Mice with the AQP11 recessive mutation C227S, an alteration that affects the maintenance of the AQP11 oligomeric structure, demonstrated higher mitochondrial injury than WT mice. Glucose treatment, which induces ROS production, increased AQP11 expression in WT mice, but this was lost in the heterozygote mice. Proximal tubule cells with the AQP11 recessive mutation respond to glucose by elevating the production of ROS, thus suggesting that AQP11 plays a protective role in glucose-induced oxidative stress in the kidney proximal tubules [51]. Although no pathological conditions have been correlated with altered AQP11 in humans, AQP11 rs2276415 variation was characterized as a genetic factor of predisposition to develop kidney diseases such as chronic kidney disease [52], transplantation failure [53], or type 2 diabetes [54]. So far, no functional assessment of the AQP11 rs2276415 variation, a missense mutation G102S, has been carried out; however, the amino acid substitution is located immediately after the first pore-forming NPC motif, suggesting the promotion of translation and folding defects, since a previous mutation from NPC to NPA led to oligomerization failure [55].

Peroxiporins also play a crucial role in maintaining lens transparency. Studies have shown that KO mice for AQP0, AQP1, and AQP5 exhibited reduced oxidative damage to the lens induced by H_2_O_2_. This indicates that these aquaporins facilitate the transport of H_2_O_2_ into the lens cells, where it is scavenged by GPX1 to maintain oxidative homeostasis. The absence of GPX1 impairs this process, leading to a decrease in H_2_O_2_ transport. The fact that GPX1 KO lenses exhibit H_2_O_2_ accumulation underscores the role of GPX1 in regulating H_2_O_2_ permeation through membranes via AQPs. Further supporting the importance of peroxiporins, an AQP5 KO hypoglycemic mouse model demonstrated significant impairment in lens transparency, accompanied by increased H_2_O_2_ production and accumulation, compared to WT mice [7]. This suggests that AQP5 is particularly critical in preventing oxidative damage and maintaining lens clarity under hypoglycemic conditions. The increased H_2_O_2_ levels in these knockout models highlight the protective role of AQPs in mitigating oxidative stress within the lens. These findings collectively unveil peroxiporins as promising new targets for the prevention or treatment of age-related lens cataracts. By modulating the activity of specific AQPs, it may be possible to develop therapeutic strategies aimed at preserving lens transparency and preventing cataract formation associated with oxidative stress.

In the human brain, AQP0, AQP1, AQP9, and AQP11 have been described in the cortex and hippocampus and their expression correlated with aging and Alzheimer’s disease grade. Experiments using 1321N1 astroglia cells and SHSY5Y differentiated neurons in inflammatory and hypoxia conditions showed upregulated AQP0, AQP1, and AQP11 transcripts. AQP0 and AQP11 were also increased at the protein level, localized in the plasma membrane and endoplasmic reticulum, respectively [56]. The peroxiporin roles were assessed by quantifying lipid peroxidation levels after the H_2_O_2_ challenge, resulting in diminished lipid peroxidation in lipopolysaccharide (LPS)-treated cells. These results highlight the potential of increasing brain peroxiporin activity to alleviate age-related injury and neurodegeneration.

Recently, experimental data revealed the AQP6 peroxiporin activity in mesothelioma cells [16]. The prolonged exposure to increased levels of ROS in tumoral environment cells boosts cell proliferation and migration and thus cancer progression, which culminates in cell death by ferroptosis. In malignant pleural mesothelioma cells confronted with heat stress, AQP6 silencing resulted in decreased H_2_O_2_ efflux and impaired cell proliferation, revealing AQP6 as an H_2_O_2_ channel and its protective feature towards ferroptosis [16]. These experiments suggest that AQP6 is a crucial player in the etiology and also the resistance to chemotherapy in mesothelioma.

AQP3 and AQP5 were shown to mediate H_2_O_2_ transport in pancreatic ductal adenocarcinoma BxPC3 cells. The lower H_2_O_2_ diffusion measured in AQP3- and AQP5-silenced pancreatic cells was correlated with reduced cell migration, which could be recovered by additional exogenous oxidative stimuli [14]. In this model, AQP5 showed a highly efficient peroxiporin activity compared to AQP3, revealing the major role of AQP5 in the dynamic fine-tuning of the endogenous concentration of H_2_O_2_ and possibly an influence on cancer initiation and progression [14]. Additionally, AQP5 peroxiporin activity was correlated with cell-cell adhesion, cell stiffness, and increased membrane permeability [57]. Moreover, AQP3 was found to be an oncogenic factor in lung adenocarcinoma, due to its peroxiporin activity and ability to increase intracellular ROS levels, essential for the etiology of the disease [58]. The study describes the inactivation of PTEN as a response to AQP3-associated intracellular ROS accumulation and subsequent activation of the AKT/mTOR pathway with consequent inhibition of autophagy and induction of cell proliferation. Altogether, these data highlight the role of peroxiporins in oxidative mechanisms underlying adenocarcinomas.

AQP8 expression was detected in both plasma and mitochondrial membranes of β-cells in the endocrine pancreas [59]. Although H_2_O_2_ regulates cell growth, in type-1 diabetes mellitus the excess of intracellular ROS accumulation in β-cells leads to the failure of insulin secretion. In RINm5F β-cells, AQP8-KO resulted in cell death, which was prevented by re-expressing AQP8 into the cells. Instead, AQP8-overexpressing cells showed higher proliferative features and increased insulin content, highlighting the role of AQP8 in H_2_O_2_ homeostasis in pancreatic β-cells and the development of type-1 diabetes mellitus [59].

Peroxiporins have emerged as promising targets for enhancing therapies, making them a focal point in redox research. The following sections explore the implications and targeting potential of peroxiporins in inflammation and cancer.

### 3.1. Peroxiporins in Inflammation

Increasing evidence demonstrates the involvement of AQPs in the inflammatory process, particularly since the activation of immune cells and the inflammatory response was described as a result of rapid cell volume changes induced by modifications in the osmotic microenvironment, with a consequent increase in cell hydraulic permeability, cell size, and cytoskeleton modifications [60].

Several peroxiporins have been detected in cells from innate and adaptative immunity [61,62], and crucial physiological roles have been demonstrated using in vitro and in vivo models. AQP5 and AQP7 are expressed in dendritic cells and participate in antigen uptake and endocytosis [63,64]. In fact, AQP3-dependent H_2_O_2_ influx may control the endocytosis process in CD8+ T cells [65]. AQP3 also participates in phagocytosis in macrophages and macropinocytosis in dendritic cells [66,67]. AQP9 was detected in the lamellipodium edges of leukocytes, suggesting an essential role in cell shape changes and mobilization upon chemokine gradients [68]. Additionally, AQP7 was shown to participate in antigen processing and migration towards a chemoattractant [64]. AQP3 and AQP9, the most relevant AQPs in the immune system, are involved in cell migration together with AQP5 [69]. AQP3 affects T cell and macrophage migration and is essential for their physiological function in system defense [70], while AQP5 and AQP9 regulate migration in neutrophils [69]. AQP1 was also correlated with cell migration and lamellipodium formation in macrophages, suggesting an influence on M0/M2 phenotype switching and motility regulation [71].

AQP3 and AQP9 implications in immune cell activation and response upon external aggression were also described [72]. The depletion of AQP9 in a murine model resulted in deficient neutrophil and T cell activation [73], while in macrophages AQP9 is a key player for the sensing system towards infection by *Pseudomonas aeruginosa* [74]. Such activation induces AQP9 upregulation with consequent changes in cell shape and motility, confirming its pivotal participation in infection, inflammation, and clearance/disease progression [74]. In vitro LPS stimulation also induces AQP1, AQP3, and AQP9 upregulation [72,75,76], and patients with inflammatory diseases show increased AQP9 in activated leukocytes [77]. In dendritic cells, AQP3, AQP5, and AQP9 are associated with cell priming [78] and proliferation [61]. In THP-1 monocytic cells, AQP3 is implicated in IL-6, pro-IL-1β, and TNFα transcription via TLR4 engagement upon LPS priming [79], and the activation of the NLRP3 inflammasome, known to be upregulated in sepsis [80], is influenced by AQP3 expression [79]. The upregulation of AQP3 might enable the well-characterized fast volume changes described to precede IL-1β secretion [81].

Sepsis is a systemic inflammatory disorder caused by a deregulated response to infection. Recently, AQP3 and AQP9, the most representative AQPs in macrophages and neutrophils, respectively, were shown to respond differently in the sepsis clinical condition. An AQP3 increase in macrophages is associated with survival, while AQP9 in neutrophils is inhibited during a prolonged inflammatory state and its expression was considered an independent risk factor for sepsis lethality [82]. Several other studies reported an altered AQP expression in sepsis patients or related complications. Leukocyte AQP1 expression was found to be increased in septic patients compared with non-septic patients in intensive care units [75]. AQP9 was augmented in neutrophils in systemic inflammatory response syndrome patients [77]. Increased AQP5 in white blood cells correlates with higher mortality [83]. Cardiac dysfunction is a common disorder in the setting of sepsis. In the LPS-induced sepsis murine model, cardiac function is impaired along with increased levels of AQP1 and proinflammatory cytokines [84].

In osteoarthritis, a degenerative disease caused mainly by chondrocyte apoptosis and cartilage matrix degradation, AQPs have been detected in cartilage cells with an impact on the regulation of cartilage physiology [85]. In fact, in Sprague Dawley rats with surgically induced osteoarthritis, AQP1 was positively correlated with caspase-3 activity, suggesting its influence on caspase-3 activation resulting in chondrocyte apoptosis [86].

In asthma, where chronic inflammation of the airways is triggered by factors including oxidative stress, AQP3 peroxiporin activity is thought to be pivotal in the initiation of the inflammatory process. In a murine model of ovalbumin-induced asthma, AQP3 associates with chemokine production (CCL24 and CCL22) in alveolar macrophages and with the transmigration of T cells [87].

In Crohn’s disease and colitis, transcellular fluid movements and water absorption are affected as a result of chronic inflammation in the colon. Apart from being crucial in water homeostasis, AQPs such as AQP3 and AQP8 are found impaired in 2,4,6-trinitrobenzene sulfonic acid (TNBS)-induced colitis and associated with increased intestinal inflammation and injury [88]. In Crohn’s disease and ulcerative colitis human biopsies, the detected downregulation of AQP8 suggests a defense mechanism against severe oxidative stress. In a murine model of 5-fluorouracil (5-FU)-induced diarrhea, AQP8 inversely correlated with pro-inflammatory cytokines (TNF-α, IL-1β, IL-6, IL-17A, and IL-22) [89]. AQP3, an important isoform in the colon regulating water fluxes through the epithelial cells, is upregulated in diarrhea and its normal function can be restored by suppressing the protein kinase A (PKA)/cyclic adenosine monophosphate response element binding protein (pCREB) signaling pathway [90].

In chronic liver injury (CLI), AQP3 is involved in macrophage priming, causing the inflammatory process. In a CCL4-induced CLI murine model, the administration of anti-AQP3 monoclonal antibody prevented H_2_O_2_ influx and cell activation, resulting in the blockage of inflammation [91].

### 3.2. Peroxiporins in Cancer Biology

Aquaporins are central players in the key processes that characterize tumorigenesis mostly by facilitating water, glycerol, and H_2_O_2_ movements between cells and/or compartments. In particular, H_2_O_2_ fluxes influence cancer progression, proliferation, and metastasis by regulating various signaling pathways. Cell proliferation, the mechanism by which a tumor grows, is intimately associated with glycerol transport by aquaglyceroporins, providing energy for cell division [92] and cell migration, favoring cell motility and metastases [93]. Moreover, by mediating H_2_O_2_ movements, peroxiporins participate in the activation of signaling pathways that initiate tumorigenesis.

AQP3 is the most frequently cancer-associated aquaporin, being abnormally expressed in different types of cancers, such as lung [94], skin [50,95], colon [96], pancreatic [97,98], gastric [99,100], and liver [101]. Experimentally modulating AQP3 affects several tumoral-related signaling pathways [102], suggesting that AQP3 targeting can prevent and treat AQP3-overexpressing tumors [103].

AQP1 is overexpressed in multiple human cancers such as those of brain [104], biliary duct [105], colon [106], bladder [107], breast [108], lung [109] and prostate [110], and it has been used as a distinctive clinical prognostic factor.

AQP5 was suggested as an enhancer of cancer cell proliferation, migration, and survival by interacting with different pathways [111,112,113]. In general, AQP5 is found upregulated in skin [114], colon [113,115], lung [116], prostate [117] and ovarian cancers [118,119,120]. Additionally, AQP7 and AQP9 are overexpressed in malignant ovarian tumor cells compared to benign tumors or normal tissue [121] and deficiently expressed in hepatocellular carcinoma associated with tumor aggressiveness [101]. AQP9 is overexpressed in glioblastoma [122,123], and its downregulation seems to prevent apoptosis, promoting invasion and epithelial-to-mesenchymal transition [124,125].

Several studies reported the involvement of AQP-facilitated H_2_O_2_ influx in cancer cell models. For instance, AQP3-mediated H_2_O_2_ transport was shown to be involved in EGF-induced cell signaling in squamous cell carcinoma (A431) and lung cancer (H1666) cell lines, and AQP3 blockage suppresses the cell response to EGF [27]. Additionally, Erk and Akt activation induced by EGF and modulated by SHP2 and/or PTEN is impaired in AQP3-silenced cells, resulting in affected cell growth and migration. Interestingly, A431-derived xenografts showed decreased tumor growth and animals lived longer [27]. In MNT-1 and A375 melanoma cells, AQP3 blockage by gold compounds or transient silencing led to decreased intracellular ROS accumulation, suppressing cell adhesion, migration, and proliferation [126].

In breast cancer cells, AQP3 is associated with the induction of CXCL12/CXCR4-dependent cell migration [127], a critical process in breast cancer progression and metastasis. Here, H_2_O_2_ produced by CXCL12-activated NOX2 enters the cell via AQP3, generating a localized increase in intracellular H_2_O_2_ to function as a second messenger to induce cell migration. H_2_O_2_ is then oxidized by PTEN and PTP1B, resulting in Akt pathway activation. Additionally, AQP3 expression was directly correlated with motility in both in vitro and in vivo models, and spontaneous metastasis of orthotopic AQP3-silenced xenografts in the lung were reduced compared to controls [127].

In pancreatic ductal adenocarcinoma cells BxPC3, AQP3 and AQP5 facilitate H_2_O_2_ influx, and silencing these homologs results in impaired cell migration [14,57]. Deficient AQP5 was associated with more apoptotic cells [57]. 

In leukemia, AQP8 expression was correlated with a VEGF capacity to trigger increased H_2_O_2_ intracellular levels. AQP8-mediated H_2_O_2_ permeation induced phosphorylation of PI3K and p38 MAPK, resulting in cell proliferation [12].

Interestingly, a study in hepatocellular carcinoma reported that AQP9 negatively impacts cell invasion by preventing the epithelial-mesenchymal transition. Moreover, AQP9 seems to suppress liver cancer stem cell stemness. AQP9 is downregulated by insulin-like growth factor 2 (IGF2), which is crucial for cancer stem cell stemness maintenance. AQP9 overexpression results in intracellular H_2_O_2_ accumulation, attenuation of β-catenin/TCF4 interaction, dissociation of β-catenin from FOXO3a (ultimately resulting in β-catenin activity inhibition), and cancer stem cell stemness suppression [128].

### 3.3. Peroxiporins in the Inflammation-to-Cancer Transformation

A prolonged inflammatory state is closely related to cancer initiation and progression [129]. Among the external factors associated with tumorigenesis, several induce oxidative stress, such as bacterial or viral infections, obesity, smoking, and alcohol consumption, increasing cancer risk and stimulating the malignant phenotype. Cancer-associated inflammation can be a result of the initial mutations that characterize tumorigenesis, consequently contributing to the malignant progression by recruiting and activating immune cells. External factors associated with inflammation-induced immunosuppression provide the optimal conditions for tumor development [129].

The gastric infection caused by *Helicobacter pylori* (*H. pylori*) has been long associated with the development of gastric carcinomas, playing a key role in its etiology [130]. Interestingly, AQP3 overexpression correlates with both *H. pylori* infection [131] and gastric carcinoma [132]. Using in vitro and in vivo models, AQP3 protein expression in gastric cancer samples was found positively correlated with *H. pylori* infection. In fact, in vitro, *H. pylori* infection promoted AQP3 upregulation by activating the extracellular signal-regulated kinase (ERK) signaling pathway, resulting in increased cell proliferation and migration. Such an interaction was corroborated in the *H. pylori*-induced rodent model of gastric cancer [131]. The study provides insights into how a non-carcinogenic factor induces AQP3 modulation and how the event can lead to tumorigenesis.

Another study reported that the regulatory mechanisms by which AQP3 is upregulated in *H. pylori*-mediated carcinogenesis are based on the activation of ROS-dependent pathways. *H. pylori* infection induces the upregulation of AQP3 through the transcriptional regulation of HIF-1α and the increase of ROS in human gastric epithelial AGS and GES-1 cells. Additionally, proinflammatory cytokines produced in response to *H. pylori* infection, such as IL-6, IL-8, and TNFα, were AQP3 level-dependent. These data uncover the initiation of carcinogenesis upon activation of the ROS–HIF-1α axis that is later exacerbated by the ROS–HIF-1α–AQP3–ROS loop [133].

Overall, by addressing the oxidative mechanisms underlying cancer progression and treatment resistance, peroxiporins strategically offer a novel and potentially effective approach to enhancing outcomes for patients with inflammation and cancer.

## 4. Peroxiporins as New Targets to Tackle Oxidative Stress-Related Pathologies

Although the involvement of AQPs in redox signaling and oxidative stress demonstrates their potential as targets for the development of innovative therapies for inflammation and cancer, the discovery of peroxiporin modulators has progressed slowly. Several molecules have been reported to affect H_2_O_2_ flux through AQPs. Table 1 compiles the modulators of peroxiporins reported in the literature, their effect on H_2_O_2_ permeability, the experimental model, and the method used in each case.

The clinically approved AQP1 inhibitor, Bacopaside II, a triterpene saponin natural compound, impaired AQP1-mediated H_2_O_2_ permeability in mice and human myocytes, attenuating cardiac hypertrophy in a murine model [134]. The organogold (III) compound Auphen, [Au(1,10-phenanthroline)Cl2]Cl, and its derivatives (C^CO^N and C^NH^N) have been shown to have a potent effect in blocking AQP3 peroxiporin activity, reducing H_2_O_2_ permeability in melanoma [126] and pancreatic ductal adenocarcinoma cells [14] and impacting various biologic processes such as cell adhesion, proliferation, and migration. Additionally, the drug-like small molecule ortho-chloride-containing compound DFP00173 is a selective and potent AQP3 peroxiporin activity modulator, constituting a very solid isoform-specific aquaglyceroporin inhibitor with experimental and molecular modeling validation [135]. Interestingly, the increase of AQP6 peroxiporin activity might be a powerful tool to fight cancer by boosting the effect of chemotherapeutic drugs. In this context, CeO_2_, Gd_2_O_3_, and Fe_3_O_4_-loaded polyacrylic-coated solid nanoparticles were found to increase the H_2_O_2_ permeability by modulating AQP3, AQP6, and AQP8, resulting in an innovative tool to control oxidative stress in cells [136]. Experimental data has also validated HgCl_2_ and AgNO_3_, two broad AQP inhibitors, as peroxiporin activity modulators, impairing H_2_O_2_ permeability via AQP1 [19], AQP3 [12,14], AQP5 [14], and AQP8 [12,137]. In contrast, H_2_O_2_ permeability via AQP6 increases upon HgCl_2_ treatment [16]. The benzothiadiazole urea-containing molecule RF03176 is an isoform-specific AQP9 inhibitor of H_2_O_2_ permeability [135]. Recently, the compound RG100204 was reported to inhibit murine AQP9 peroxiporin activity in a CHO cell expression system [138]. Additionally, biologics are rising as safer and more selective modulators. An antibody raised against AQP3 was shown to be effective in blocking H_2_O_2_ permeability and preventing liver injury, revealing a promising innovative therapy for macrophage-dependent liver injury [91].

In conclusion, the discovery and validation of peroxiporin activity modulators and the design of new molecules with improved potency and selectivity are seen as a positive step forward in the development of innovative therapies for oxidative stress-associated disorders.

**Table 1 ijms-25-08381-t001:** Modulators of aquaporin-peroxiporin activity.

Aquaporin	Modulator	Effect on H_2_O_2_ Permeability	Cell Type/Model	Method	Reference
AQP1	Bacopaside II	Decrease	Cardiac myocytes	H_2_DCF-DA	[134]
	HgCl_2_	Decrease	Liposomes	HRP/Amplex Red	[19]
AQP3	Antibody anti-AQP3	Decrease	CHO-K1 cells	CM-H_2_DCFDA	[91]
	Au (III) C^CO^N and C^NH^N	Decrease	Melanoma cells (A375, MNT1)	H_2_DCF-DA	[126]
	Auphen	Decrease	PDAC cells (BxPC3)	H_2_DCF-DA	[14]
	HgCl_2_	Decrease	PDAC cells (BxPC3)	H_2_DCF-DA	[14]
	DFP00173	Decrease	CHO cells	HyPer-3	[135]
	AgNO_3_	Decrease	B1647 cells	H_2_DCF-DA	[12]
	Fe_3_O_4_-, Gd_2_O_3_-, and CeO_2_-loaded nanoparticles	Increase	Hela cells	HyPer7-NES	[136,139]
AQP5	HgCl_2_	Decrease	PDAC cells (BxPC3)	H_2_DCF-DA	[14]
AQP6	CeO_2_-loaded nanoparticles	Increase	Hela cells	HyPer7-NES	[136]
	HgCl_2_	Increase	Mesothelial cells	H_2_DCF-DA/HyPer7	[16]
AQP8	AgNO_3_	Decrease	B1647 cells	H_2_DCF-DA	[12]
	Fe_3_O_4_-, Gd_2_O_3_-, and CeO_2_-loaded nanoparticles	Increase	Hela cells	HyPer7-NES	[136,139]
	HgCl_2_	Decrease	Sperm cells	H_2_DCF-DA	[137]
AQP9	RF03176	Decrease	CHO cells	HyPer-3	[135]
	RG100204	Decrease	CHO cells	HyPer-3	[138]

## 5. Final Remarks

Oxidative stress results from dysbalanced redox signaling, affecting human physiology and influencing several pathological conditions such as systemic inflammation and cancer. Although recent advances in diagnostic methods and care treatments improved the life quality and longevity of patients with oxidative stress-related pathologies, the search and development of more effective diagnostics and therapies are still paramount. Growing evidence highlights peroxiporins as central players in redox signaling, since alterations in their expression and activity are correlated with ROS accumulation, the initiation of the inflammatory process, and the induction of tumor cell phenotypes. Such unique features underscore the potential of peroxiporins as both biomarkers for disease and targets for therapeutic intervention. Further research into the specific mechanisms by which these aquaporins influence disease progression is paramount, validating their importance in medical science and the potential for developing new treatment strategies.

## Figures and Tables

**Figure 1 ijms-25-08381-f001:**
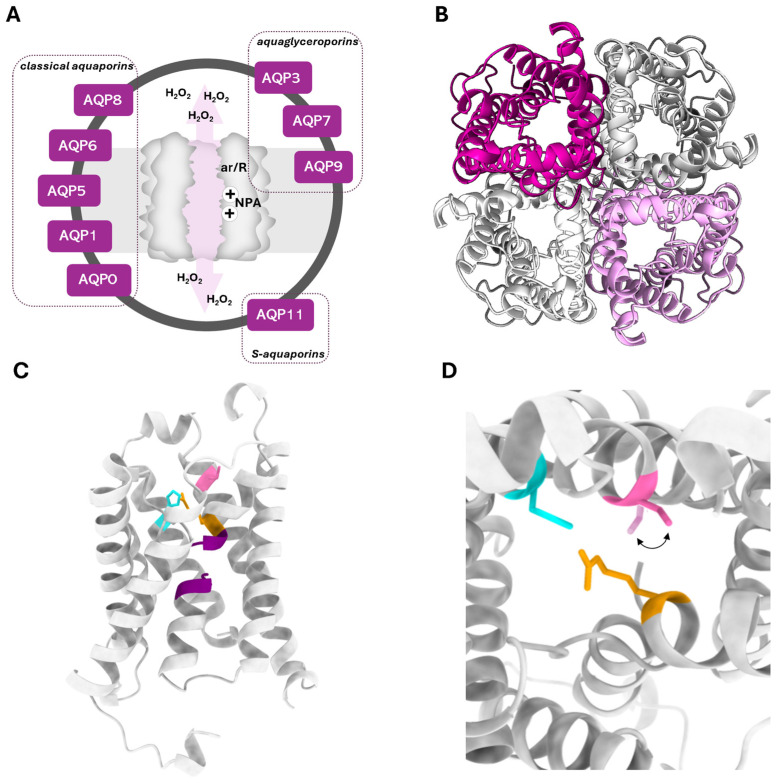
Peroxiporin features. (**A**) Graphical illustration of a peroxiporin channel displaying the two selectivity filters (size ar/R and charge NPA), and the human AQP homologues known to permeate hydrogen peroxide (H_2_O_2_). (**B**) Top view of the homotetrameric representation of the peroxiporin AQP8. (**C**) Side view of the AQP8 monomer with the two conserved asparagine–proline–alanine NPA motifs (purple colored) and cysteine–histidine–arginine arrangement (pink, turquoise, and orange colored, respectively) based on a predictive structure generated in Phyre2 web portal. (**D**) Detail of the monomer side view. The arrow indicates the conformational change induced by C53 persulfidation changing the shape of the (ar/R) filter, likely causing the aromatic ring of H72 to control the gating of AQP8. AQP8 predictive structure was generated using the Phyre2 web portal. Final figures were generated using UCSF Chimera version 1.15 software.

**Figure 2 ijms-25-08381-f002:**
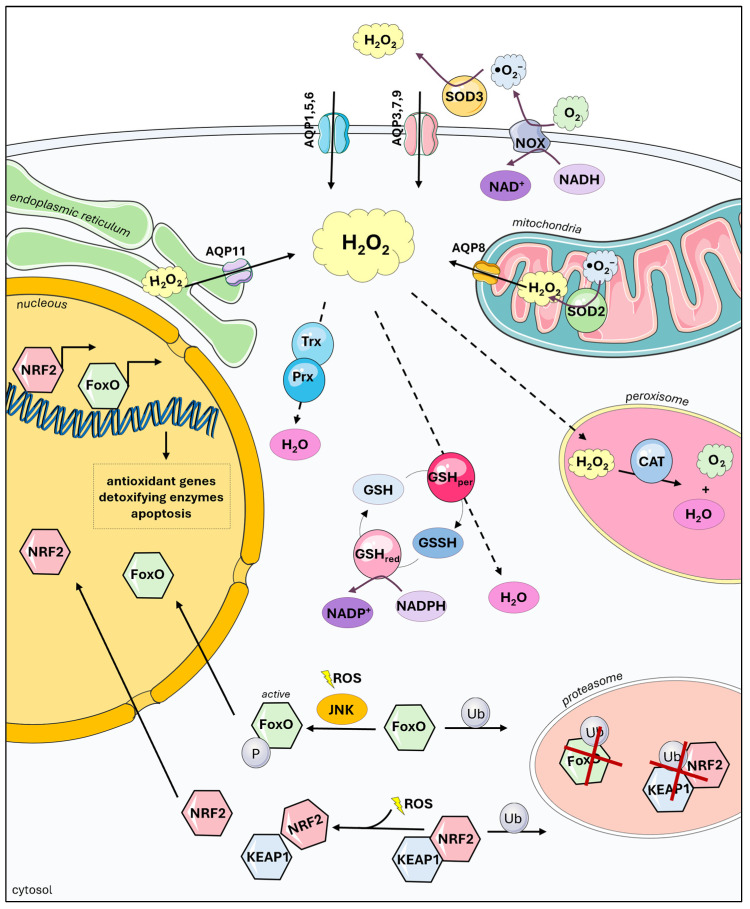
Maintenance of cellular redox homeostasis. Hydrogen peroxide (H_2_O_2_) is formed under physiological conditions in the cell by the activity of NADPH oxidases (NOX), by “leakage” of the electron transport chain in the mitochondria, and also in the endoplasmic reticulum (ER). Its fate is determined by the nearest molecules, as it can be: (1) exported by peroxiporins or (2) processed by detoxifying enzymes such as thioredoxins (Trx), peroxiredoxins (Prx), or glutathione (GSH) and its enzymatic cascade, which are under the control of the redox-sensitive transcription factors, such as the NRF2 and FOXO family.

## Data Availability

Not applicable.

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
