# Peer review of "Peroxiporins and Oxidative Stress: Promising Targets to Tackle Inflammation and Cancer"

_ijms, 2024, doi:10.3390/ijms25158381_

Round 1

Reviewer 1 Report

Comments and Suggestions for Authors

The topic of the manuscript is of relevance and general interest to the readers.  Overall, the manuscript is very well written and the authors performed a thorough, detailed, and focused review of the literature.

Recommendations:

1) Page 2, line 77: Consider revising the sentence.  "It is reasonable to admit" should be revised to "It is reasonable to assume" or similar.

2) Page 2, line 78: It seems that the word "are" between "molecules" and "transported" is missing.

3) Considering the title, it is recommended that an introductory paragraph should be added to discuss the role of H2O2 in cancer and oxidative stress, as a final paragraph in the Introduction, to orient the reader.

4) Page 3, line 94: Correct the sentence.  It seems that the words "have been" are missing after "metabolism".  Change 'Showed" to 'shown" to read: "Oxygen and oxidative metabolism have been shown to be strategic advantages...".

5) page 4, line 120.  above-mentioned is hyphenated.

6) some abbreviations are spelled out twice, e.g. SOD (page 4, lines 108 and 139).

Comments on the Quality of English Language

Minor revisions.

Author Response

Dear Reviewer, thank you for your positive comments. Please find below the answers to your questions. We hope to have clarified all the raised queries.

The topic of the manuscript is of relevance and general interest to the readers.  Overall, the manuscript is very well written and the authors performed a thorough, detailed, and focused review of the literature.

Recommendations:

1) Page 2, line 77: Consider revising the sentence.  "It is reasonable to admit" should be revised to "It is reasonable to assume" or similar.

Reply1: Thank you for the suggestion, the sentence was revised accordingly.

2) Page 2, line 78: It seems that the word "are" between "molecules" and "transported" is missing.

Reply2: The sentence now reads: “Hâ‚‚Oâ‚‚ molecules when transported through AQP8 sulfenylate C53”

3) Considering the title, it is recommended that an introductory paragraph should be added to discuss the role of H2O2 in cancer and oxidative stress, as a final paragraph in the Introduction, to orient the reader.

Reply3: A new paragraph was inserted at the end of the introduction to contextualize the aim of the manuscript.

4) Page 3, line 94: Correct the sentence.  It seems that the words "have been" are missing after "metabolism".  Change 'Showed" to 'shown" to read: "Oxygen and oxidative metabolism have been shown to be strategic advantages...".

Reply4: The sentence was amended according to the reviewer suggestion.

5) page 4, line 120.  above-mentioned is hyphenated.

Reply5: The wording was updated.

6) some abbreviations are spelled out twice, e.g. SOD (page 4, lines 108 and 139).

Reply6: The abbreviation SOD was checked in the spotted lines.

Reviewer 2 Report

Comments and Suggestions for Authors

The review has the potential to be an excellent and timely synopsis of the field, but would benefit from the addition of clear informative statements which set the perspective and built a coherent set of take-home points. This would be an excellent opportunity to present the current state of the field with future opportunities and challenges.

As presented, the review is an exhaustive summary of compiled details. Although a great starting point in terms of the effort clearly invested, it doesn't yet take advantage of the opportunity to shape research thinking in this important field of work.

For example in Section 2, there is a tendency to list long sets of signaling components without interpreting the differences in location, regulation and functional roles that would lead to testable hypotheses. For example in lines 157-159, it would be helpful to clarify the nature of the 'exploitation' being done by cancer cells, to more fully explain the benefit conferred and how cancer cells escape damage at high doses, and to define the word "normal" as set in quotation marks.

Similarly in Section 3, a substantial amount of good information has been assembled, and could provide a valuable resource for readers, especially if the details could be framed as evidence supporting a logical series of main points which remain to be added.

Comments on Figure 1: Panel A-- AQP0 is also a peroxiporin that could be added to the set. Panel B-- Protein database identifiers are needed in the legend for the structures shown. Panels B and C-- These well used views don't seem to provide any novel insights into peroxide transporting features, and might be more informative if presenting comparisons between peroxiporin and non-peroxiporin structures. Panel D-- Can a H2O2 molecule be modeled into the pore structure in panel D to show how the conformational change induces a suitable pathway?

Points in the Introduction to be addressed:

Lines 46-47. AQP0 also should be included as an anion transporting AQP, citing work of J Hall and colleagues. To be scientifically objective in your presentation, if AQP-mediated ion conductances are going to be cited, then cation transporting AQPs also merit inclusion in the list.

Line 48. Define the abbreviation "S-aquaporin"

Line 50. References 2 and 3 alone do not adequately cover the full range of results being summarized in the associated sentence.

Author Response

Dear Reviewer, thank you for your comments. Please find below the answers to your questions. We hope to have clarified all the raised queries.

[Reviewer Comments]:  The review has the potential to be an excellent and timely synopsis of the field, but would benefit from the addition of clear informative statements which set the perspective and built a coherent set of take-home points. This would be an excellent opportunity to present the current state of the field with future opportunities and challenges. As presented, the review is an exhaustive summary of compiled details. Although a great starting point in terms of the effort clearly invested, it doesn't yet take advantage of the opportunity to shape research thinking in this important field of work.

1) For example in Section 2, there is a tendency to list long sets of signaling components without interpreting the differences in location, regulation and functional roles that would lead to testable hypotheses. For example in lines 157-159, it would be helpful to clarify the nature of the 'exploitation' being done by cancer cells, to more fully explain the benefit conferred and how cancer cells escape damage at high doses, and to define the word "normal" as set in quotation marks.

Reply1: We thank the Reviewer for the suggestion. We understand that some statements need clarification. Therefore, we replaced the sentence mentioning „exploitation“ with a more specific explanation. Additionally, we provided localization details for some of the mentioned proteins. Redox pathways are very complex and intertwining, and we aimed to provide an overview of these pathways to emphasize the need to study aquaporins (specifically peroxiporins) as regulators of H2O2 flux.  We hope these revisions address the Reviewer's concerns and enhance the clarity of our manuscript.

2) Similarly in Section 3, a substantial amount of good information has been assembled, and could provide a valuable resource for readers, especially if the details could be framed as evidence supporting a logical series of main points which remain to be added.

Reply2: We appreciate the reviewer's constructive analysis of the manuscript. Section 3 discusses the implications of peroxiporins in oxidative stress and signaling pathways that may affect disease outcomes. The text has been restructured to highlight the involvement of each AQP in disease outcomes and their associated pathways.

3) Comments on Figure 1: Panel A-- AQP0 is also a peroxiporin that could be added to the set. Panel B-- Protein database identifiers are needed in the legend for the structures shown. Panels B and C-- These well used views don't seem to provide any novel insights into peroxide transporting features, and might be more informative if presenting comparisons between peroxiporin and non-peroxiporin structures. Panel D-- Can a H2O2 molecule be modeled into the pore structure in panel D to show how the conformational change induces a suitable pathway?

Reply3: We apologize for the mistake and thank the reviewer for spotting this gap in Fig 1. Panel A was updated to include AQP0 as a peroxiporin; the related text was also updated, and an additional reference was inserted. Panel B: The structure of AQP8 is not available in the literature. An AQP8 predictive structure was generated based on the amino acid primary sequence using the Phyre2 web portal, and the final figures were generated with UCSF ChimeraX software. Panel B,C and D were created to illustrate the mechanism reported in the literature of H2O2 transport through AQP8 and highlight the crucial residues prone to regulate H2O2 permeation (Bestetti et al 2018. Sci Adv, 4(5), eaar5770), as an example.

In this review paper we did not aim to perform original computational studies; modeling H2O2 molecules inside the channel and comparing peroxiporin and non-peroxiporin structures are novel approaches that were not yet explored and might be developed in future work.

Points in the Introduction to be addressed:

4) Lines 46-47. AQP0 also should be included as an anion transporting AQP, citing work of J Hall and colleagues. To be scientifically objective in your presentation, if AQP-mediated ion conductances are going to be cited, then cation transporting AQPs also merit inclusion in the list.

Reply4: The specific permeability features of AQP0, as well as AQP-mediated ion conductances, are out of the scope of this paper. However, a sentence was inserted in the introduction stating that, in addition to the monomeric pore, a central pore in the tetrameric protein can serve as a pathway for the flux of gases or charged particles, such as ions.

5) Line 48. Define the abbreviation "S-aquaporin"

Reply5: The term “S-aquaporin” was explained in the context.

6) Line 50. References 2 and 3 alone do not adequately cover the full range of results being summarized in the associated sentence.

Reply6: The references have been updated to support the roles of aquaporins.

Round 2

Reviewer 2 Report

Comments and Suggestions for Authors

The authors have clearly addressed the concerns raised. The perspective statements and interpretive summaries are excellent additions to the MS that will boost accessibility and general interest for a broad audience.